# Safe Reuse of Wastewater: Organic Contaminants Degradation and Sanitization by Ozone in a Modulable Continuous-Flow System

Biagio Esposito [1], Massimo Capobianco [2], Maria Luisa Navacchia [2], Gianpaolo Sabia [3], Roberta Guzzinati [3], Francesco Riminucci [1,4], Luca Bolelli [5], Federico Ponti [6], Giorgio Longino [2] and Elida N. Ferri [5,*]

[1]  Proambiente S.c.r.l., Tecnopolo Bologna, National Research Council CNR, Via P. Gobetti 101, 40129 Bologna, Italy; b.esposito@consorzioproambiente.it (B.E.); r.riminucci@consorzioproambiente.it (F.R.)
[2]  Institute for Organic Synthesis and Photoreactivity (ISOF), National Research Council of Italy (CNR), Via P. Gobetti 101, 40129 Bologna, Italy; massimo.capobianco@isof.cnr.it (M.C.); marialuisa.navacchia@isof.cnr.it (M.L.N.); giorgio.longino@isof.cnr.it (G.L.)
[3]  ENEA—LEA Laboratory for the Environment, Via Martiri di Monte Sole 4, 40129 Bologna, Italy; gianpaolo.sabia@enea.it (G.S.); roberta.guzzinati@enea.it (R.G.)
[4]  Institute of Marine Science (ISMAR), National Research Council (CNR), Via P. Gobetti 101, 40129 Bologna, Italy
[5]  Department Pharmacy and Biotechnology, University of Bologna, Via S. Donato 15, 40127 Bologna, Italy; luca.bolelli@unibo.it
[6]  Medical Equipment Technologies srl (MET), Via Palazzetti 26, 40068 Bologna, Italy; direzione@o3met.com
*   Correspondence: elidanora.ferri@unibo.it

**Featured Application: The device is envisaged as a flexible tool with which to degrade organic compounds and sanitize various types of contaminated water, even on-site.**

**Abstract:** Effective treatments improving both the chemical and microbiological quality of reclaimed wastewater are urgently needed. Ozone is a clean, economic, and environmentally friendly method to sanitize solutions and surfaces and to degrade organic pollutants. A simple, continuous-flow water-ozoniser system was tested to evaluate its effectiveness in batch treating various kinds of wastewater, including the effluent from small municipal plants. The degradation effects on a mixture of urban and industrial standard pollutants were investigated by HPLC-UV-MS analysis and biotoxicological assays. The results revealed that the concentration of most organic pollutants was reduced to 20–0% of the initial one within one hour. One resultant compound was recalcitrant (40% reduction only). The bioassays indicated the definitive reduction in toxic effects after treatment. Similar results were obtained when secondary, post sedimentation, wastewater treatment plant effluents were treated. Heterotrophic plate counts confirmed the strong biocidal activity of ozone. The developed prototype can successfully treat locally produced wastewater, secondary effluents from small–medium plants, and non-potable water resources.

**Keywords:** wastewater reuse; ozone; AOPs; sanitization; recirculation method

## 1. Introduction

Recycling and reusing any kind and volume of wastewater (WW) seems to be one of the immediately feasible ways to cope with water scarcity. However, in order to provide safe water for human consumption the disinfection and removal of pollutants are mandatory to avoid undesirable ecological and human health effects. Additionally, wastewater of different quality and origin, surface water, and groundwater can be equally contaminated by microorganisms and chemical compounds, thus always requiring adequate treatment before being considered safe for human consumption [1–3].

Despite their low concentration ($\mu g \cdot L^{-1}$ to $ng \cdot L^{-1}$), micropollutants, and in particular the emerging pollutants, are objects of great concern because of their toxic, genotoxic, and endocrine disrupting effects. Pharmaceuticals, endocrine disruptors, personal care products, pesticides, industrial dyes, and polycyclic aromatic hydrocarbons contaminate both urban and industrial wastewater since they are released into the environment mostly unchanged in their structure and activity, are resistant to biodegradation, and accumulate in water bodies, crops, and soil [4,5]. Wastewater treatment plants (WWTPs) are not designed to remove these pollutants; thus, new effective treatments are urgently needed and some of them had already been implemented.

Chemical disinfection procedures based on chlorine compounds can give rise to disinfection by-products, some of which are carcinogenic in humans and toxic to the environment [6,7]. Moreover, some pathogenic organisms are not inactivated by these treatments [8,9] and this limits the reuse of the treated effluents [10].

For decades, advanced oxidation processes (AOPs) [11], based on fast radical reactions, have been tested as an alternative to, or have been used in combination with, classical WW treatment methods exploiting their disinfectant and oxidative degradation abilities [12–18]. Among them, ozone has demonstrated an ability to act as an effective parasiticide, germicide, and virucidal agent [19–21], ensuring the sanitization of several fresh and dry foods, [22–27] working places, and surfaces. As far as water is concerned, ozone treatment improves its quality, removing the color, smell, and taste at typical disinfection doses [28]; oxidizing inorganic compounds (arsenic, iron, manganese) to form insoluble substances; breaking down quickly without leaving residues or toxic compounds; and enhancing the biodegradability of organic pollutants in wastewater [29]. These features further indicate that ozone is the most appropriate approach in several particular cases [30–35].

In various countries, wastewater utilities have installed or are in the process of installing ozone generators to treat secondary effluents [36,37]. Potable water utilities are doing the same to remove a variety of contaminants in potable water, surface water, and groundwater [1,19].

However, wastewater not reaching the treatment plants or drinking water not coming from a suitable potable water utility are a common reality, even in developed countries, requiring effective and flexible solutions.

The present study aimed to investigate the performance of a self-assembled new prototype of a ozone treatment system which coupled the generator with a recirculation system to create a continuous flow of water and a repeated injection of ozone. The rate of pollutant degradation was determined by HPLC-UV-MS, the variation in toxicity by biotoxicological tests and the sanitization capacity by heterotrophic plate count.

These results should demonstrate the capability of the system in reducing biotoxicity and in sanitizing biologically contaminated water, indicating that the prototype is a versatile tool for water reclamation, especially in locations without or far from wastewater and potable water utilities.

## 2. Materials and Methods

### 2.1. Experimental Set-Up

To evaluate the capability of the system to degrade organic pollutants in complex solutions, a mixture of six compounds which represent the most common classes of pollutants was prepared at a concentration higher than usually detected in real wastewater in order to follow the oxidative degradation by HPLC-UV-MS assay. Amounts of 25 mg of each compound were dissolved in 250 mL of methanol and the solution was treated in an ultrasound bath (Bransonic 5, Cecchinato A. sas, Mestre, Italy) for 15 min. An amount of 100 μL of the solution was diluted 1:10 with distilled water, and starting from this solution, the calibration curves for the HPLC measurements were prepared. The remaining volume was diluted in 50 L of tap water.

Samples of WWTP secondary effluent were kindly supplied by the WW utility manager HERA SpA (Bologna, Italy), collected at the municipal WWTP of Cesena, (Italy). The

WWTP (about 195,000 Inhabitant Equivalent) treats domestic contributions from Cesena and Cesenatico cities, industrial discharges, and sewage trucks. Volumes of 50 L were collected in a sterilized container at the outlet of the sedimentation tanks, refrigerated, and stored at 4 °C until the experimental treatments. The samples appeared clear, without suspended solids.

### 2.2. Chemicals and Reagents

The compounds selected to prepare the solution in methanol were: Imidacloprid (insecticide), benzophenone-4 (sunscreen in personal care products), Bisphenol-A (plasticizer), Carbamazepine (anticonvulsant), Cyprodinil (fungicide), and Ofloxacin (fluoroquinolone antibiotic). Tap water was chosen as the solvent. The compounds and the methanol were supplied by Sigma Aldrich (Milano, Italy). To perform the HPLC-UV-MS analysis ammonium acetate, formic acid, and LC-MS grade acetonitrile at the highest available purity degree were purchased from Sigma-Aldrich and used without any further purification. The components of the *Vibrio fisheri* nutrient broth (NaCl 15 g, peptone 2.5 g, yeast extract 1.5 g, glycerol 1.5 mL, HEPES 0.01 M in 500 mL, pH 7) were supplied by Sigma-Aldrich, as well as the salts to prepare the Jaworski's culture medium: $Ca(NO_3)_2 \cdot 4H_2O$ 20 g $L^{-1}$; $KH_2PO_4$ 12.4 g $L^{-1}$; $MgSO_4 \cdot 7H_2O$ 50 g $L^{-1}$; $NaHCO_3$ 15.9 g $L^{-1}$; EDTAFeNa and $EDTANa_2$ both at 2.25 g $L^{-1}$; $H_3BO_3$ 2.48 g $L^{-1}$; $[(NH_4)_6Mo_7O_{24} \cdot 4H_2O]$ 1 g $L^{-1}$; $MnCl_2 \cdot 4H_2O$ 1.4 g $L^{-1}$, cyanocobalamin, biotin, and thiamine, each one 0.04 g $L^{-1}$, $NaNO_3$ 80 g $L^{-1}$; $NaH_2PO_4 \cdot 2H_2O$ 36 g $L^{-1}$ in distilled water and the f/2 medium: $EDTANa_2$ 4.16 g $L^{-1}$, $FeCl_2$ $6H_2O$ 3.15 g $L^{-1}$, $CuSO_4$ 5 $H_2O$ 0.01 g $L^{-1}$, ZnSO4 $7H_2O$ 0.022 g $L^{-1}$, $CoCl_2$ $6H_2O$ 0.01 g $L^{-1}$, $MnCl_2 \cdot 4H_2O$ 0.18 g $L^{-1}$, $Na_2MoO_4$ $2H_2O$ 0.006 g $L^{-1}$, Cyanocobalamin and biotin, each one 0.0005 g $L^{-1}$, Thiamine HCl 0.1g $L^{-1}$, $NaNO_3$ 0.075 g $L^{-1}$, $NaH_2PO_4 \cdot 2H_2O$ 0.0056g $L^{-1}$ in artificial seawater (Instant Ocean Sea salts, from Instant Ocean, Blacksburg, VA, USA), the growth medium of the alga *Raphidocelis subcapitata* and *Dunaliella tertiolecta*, respectively. The non-selective solid medium employed in the heterotrophic plate count assay was the "Plate Count Agar" (Casein-peptone, Dextrose, Yeast Agar) from Sigma-Aldrich.

### 2.3. The Ozone Treatment System

The system was quite simple, very flexible in its structure, easy to displace, and suitable to work on site. Currently, the generator can be employed with and without a hydraulic system, to treat seasoning or storage cells, food washing lines, toilets, showers, and swimming pool water, respectively. The ozone generator "Poolsan" and the hydraulic system performing the continuous water flow treatment were designed and assembled at the company, MET SrL (Bologna, Italy). The ozone generator was available in two models, producing 2 or 4 g/h, with a power consumption of 300 or 400 Watt/h, respectively. Pure gaseous ozone was produced by an electric discharge (7000–15,000 V) in a pure oxygen atmosphere obtained from compressed and molecular sieve-filtered air. A Venturi jet mixer continuously fed the liquid flow with the ozone microbubbles stream. The liquid flow was conveyed to a blender tower designed to improve the water-ozone mixing; the blender was divided into two inner compartments and it functioned by converse flows supported by a recirculation pump (delivery of 40L $min^{-1}$). The volume of the blender was defined by the volume of water under treatment. In our experiments it was 1 L, suitable to treat 50 L of water. The concentration of dissolved ozone, 4 ppm, was measured by an analytical kit based on the indigo method [38] and a photometer, both provided by Hach-Lange (Milano, Italy). The generator was connected to a computer network for remote control.

The "Poolsan" generator and most parts of the hydraulic system are shown in Figure 1.

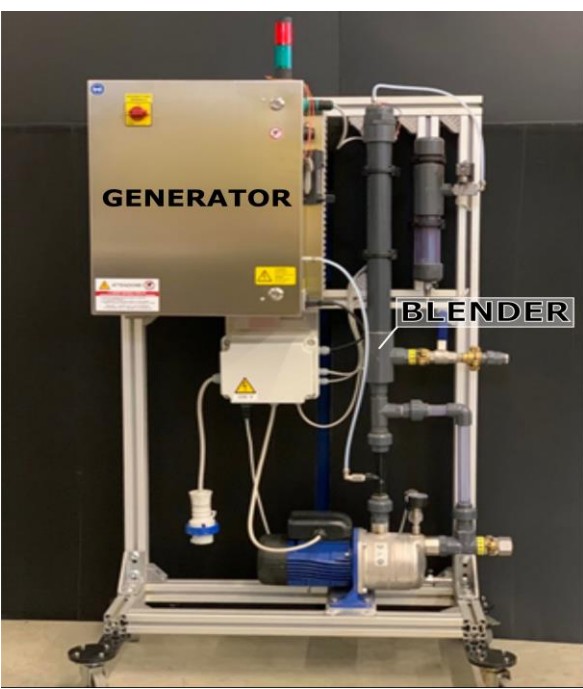

**Figure 1.** The waters ozone treatment system assembled at MET, based on the ozone generator "Poolsan" and hydraulic components to perform the recirculation procedure. The generator and the main component of the hydraulic system are mounted on a mobile support.

### 2.4. The HPLC-UV-MS Analyses

The untreated and treated mixture samples collected at different sampling times were analyzed by HPLC-DAD-MS. The analyses were performed on a Dyonex Ultimate 3000 HPLC (Thermo Fisher Scientific, Milano, Italy) equipped with a diode array UV and a mass spectrometer TSQ Quantum Access Max with an electrospray ionization source detector. 0.5 mL samples were used for the automated injection. The chromatographic separation was performed on a reverse phase Zorbax SB-C18 column 4.6 × 150 mm, 5 microns (Agilent Technologies Italia, Cernusco sul Naviglio, Italy), at a flow rate of 0.5 mL/min. Details of the HPLC and UV-MS analysis conditions were reported in Tables 1 and 2, respectively.

**Table 1.** Chromatographic separation details.

| Time (min) | A: $NH_4OAc$ 5 mM in Water/0.01% HCOOH | B: ACN | C: $H_2O$ |
|:---:|:---:|:---:|:---:|
| 0 | 98 | 2 | 0 |
| 14 | 5 | 95 | 0 |
| 15 | 0 | 95 | 5 |
| 35 | 0 | 95 | 5 |
| 40 | 98 | 2 | 0 |

The degradation degree was calculated by using the respective calibration curves, and the limit of detection (LOD) of each analyte was established as the first lowest calibration point. The analysis were performed in duplicate.

**Table 2.** UV-MS analysis details.

| Compound | $\lambda_{max}$ (nm) | ESI Parent Ion | Product Ion | Collision Energy |
|---|---|---|---|---|
| OFLOX | 296 | [M + 1] 362.000 | 318.000 | 20 |
| IMID | 220 | [M + 1] 256.000 | 208.900 | 16 |
| BP-4 | 282 | [M − 1] 307.000 | 226.900 | 30 |
| CBZ | 285 | [M + 1] 237.000 | 193.900 | 18 |
| BPA | 275 | [M − 1] 227.000 | 212.200 | 20 |
| CPD | 260 | [M + 1] 226.000 | 93.000 | 34 |

*2.5. The Samples Organic Load*

The organic load in the WWTP effluent samples was evaluated spectrophotometrically by determining the COD and the TOC values by using the Hach LCK 314 Cuvette Test (15–150 mg/L $O_2$) and the TOC Hach-Lange test kit, following the manufacturer's instructions, respectively. Measurements were conducted using a DR5000 spectrophotometer. The above-mentioned kits and the spectrophotometer were obtained from Hach-Lange, Milano, Italy.

*2.6. The Biotoxicity Assays*

2.6.1. The Bioluminescent Bacteria Light Emission Inhibition Assay

Lyophilized aliquots of luminescent bacteria *V. fischeri* cultures were prepared at our laboratory starting from fresh cultures originating from a batch supplied by the Pasteur Institute (Paris, France). The 96-well "Black Cliniplate" microplates were supplied by Thermo Scientific (Vantaa, Finland) and the luminometer was the Victor Light 1420 model from PerkinElmer, USA.

According to [39], lyophilized aliquots of *V. fischeri* were reconstituted with 1 mL of distilled water and re-suspended in 10–30 mL of nutrient broth. To treated and untreated samples was added NaCl to reach the concentration of 3% *w/v*. 200 μL of the bacteria suspension and 100 μL of each sample were dispensed into the microplate wells. The controls consisted of 200 μL of bacteria plus 100 μL of a 3% NaCl solution in tap water. The emitted light was recorded at fixed intervals in the range 0–48 h. For each sample, 5–12 replicates were prepared and the light emission intensity, reported as relative luminescence units (RLU), was expressed as mean ± SD. Moreover, the bioluminescence inhibition percentage (*I*%) was used to express the toxicity of the samples and calculated according to:

$$I\% = \frac{L\ blank - L\ sample}{L\ blank} * 100 \qquad (1)$$

where *L* is the light emitted intensity of the *sample* or of the control *(blank)*.

2.6.2. The Algal Growth Inhibition Assay

The freshwater microalga *Raphidocelis subcapitata* (previously *Pseudokirchneriella subcapitata*) and the marine green alga *Dunaliella tertiolecta* cultures were supplied by The Istituto Zooprofilattico Sperimentale of Abruzzo and Molise "G. Caporale" (Teramo, Italy). The assays were performed according to [40] with minor modification concerning the evaluation of the results; the effects on the growth were determined after 7–10 days from the beginning of the test.

The starter culture of *Raphidocelis subcapitata* was prepared by inoculating in Erlenmeyer flasks with 1 mL of microalgae suspension per 100 mL of the Jaworski's culture medium. The flasks were illuminated by a white lamp/red lamp Osram daylight $2 \times 36$ W plus an Osram Gro-Lux lamp 36 W (Osram, Milano, Italy) following the cycle: 8 h light/16 h dark, at room temperature (20 °C). To start the tests, flasks were filled with treated or untreated samples added with a suitable amount of Jaworski's medium salts and diluted algal suspension (approximately $10^5$ cells mL$^{-1}$), kept in the same conditions of the starting

culture. Controls were prepared by adding algae suspension to the Jaworski's salts mixture. The algal growth was evaluated by measuring the absorbance at 663 nm, according to a standardized indirect method for cell counting [41]. Aliquots of carefully hand shaken samples or controls were measured, in triplicate, and then the aliquots were poured back into the flask.

In parallel, the starter culture of *Dunaliella tertiolecta* was prepared by inoculating in Erlenmeyer flasks 1 mL of microalgae suspension per 100 mL of f/2 medium. The tests were performed exactly as for *R. subcapitata*.

### 2.7. The Heterotrophic Plate Count

The Petri dishes, $\varnothing$ 9 cm, for the heterotrophic plate counts (HPC) were supplied by Nuova Aptaca srl (Asti, Italy). Different volumes of ozone treated and untreated effluent samples, in the range 100–1000 µL, were distributed in the plates in triplicate. The solid medium solution was sterilized and poured into the plates when its temperature was about 37 °C. In this way, the samples were perfectly included in the medium, still liquid at that temperature. Incubation was performed at 37 °C for 24–48 h and then the number of colony-forming units (CFU) was determined. A regrowth test was carried out on specimens of the treated samples stored in stopped, sterile vials for 7 days at room temperature. These specimens were plated and the CFUs counted as described above.

## 3. Results

### 3.1. Samples

The mixture prepared to evaluate the degradation of pollutants by HPLC-UV-MS assay was stored in the dark at room temperature, and employed without any further manipulation after the ozone treatment. The effluent samples were stored refrigerated both before and after the treatment. Negligible or no changes were detected in the pH values before, during, and after the treatment and this parameter ensured ozone solution stability. The pH value was determined also in the case of the WWTP samples. A moderate pH increase was observed as treatment time increased. In Table 3, the mean values of different experiments for both kinds of sample are reported.

**Table 3.** pH values of the pollutants' mixture and WWTP samples before and after the various treatment intervals. Mean values of $n = 4$ experiments.

| | pH Value $\pm$ Uncertainty | | |
|---|---|---|---|
| Time Interval (h) | Mixture | Time Interval (h) | WWTP Samples |
| 0 (before treatment) | 7.87 $\pm$ 0.11 | 0 (before treatment) | 7.43 $\pm$ 0.10 |
| 1 | 7.66 $\pm$ 0.25 | 15″ | 7.63 $\pm$ 0.15 |
| 2 | 7.89 $\pm$ 0.38 | 30″ | 7.75 $\pm$ 0.21 |
| 4 | 7.96 $\pm$ 0.40 | 1 | 7.78 $\pm$ 0.25 |
| 6 | 7.98 $\pm$ 0.33 | 2 | 7.92 $\pm$ 0.09 |
| 8 | 8.06 $\pm$ 0.21 | 4 | 7.93 $\pm$ 0.21 |
| Tap water | 7.80 $\pm$ 0.21 | 6 | 8.02 $\pm$ 0.05 |

### 3.2. The Ozone Degradation of Contaminants

The mixture of the six pollutants was bubbled with ozone continuously for 8 h. Samples were withdrawn during the treatment after 1, 2, 4, 6 h, and again at the end of the treatment. Such a long treatment time, with respect to the usual ozone treatments, was employed because of our aim, as mentioned at the beginning of this work, to obtain information on the system's performance in the same conditions of a previously developed device [42]. Moreover, not all organic contaminants are easily degraded by ozone or hydroxyl radicals in a short time. The molecular structure of the compounds in the mixture, shown in Figure 2, was suitable to be attacked by ozone thanks to the presence of several double bonds, but this characteristic is not always a guarantee of rapid degradation.

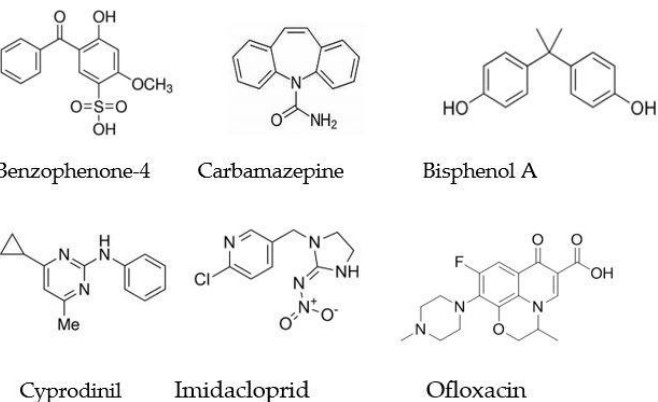

**Figure 2.** The chemical structure of the active principles mixed in solution to test the degradation power of the ozone treatment.

A different sampling schedule was also employed during the treatment of effluent samples: specimens were collected after 15 min, 30 min, and 1, 2, 4, and 6 h, to evaluate the sanitizing effects after a time shorter than 1 h.

### 3.2.1. The HPLC-DAD-MS Analysis

The mixture specimens collected after 1, 2, 4, 6, and 8 h of treatment were analyzed using HPLC-UV-MS to evaluate the parent molecules' disappearance. No detailed analysis of the degradation products of each compound was performed since this procedure was out of the scope of this work

The chromatographic peak positions of each mixture component obtained via the HPLC-UV-MS analysis of the untreated mixture were collected in Figure 3.

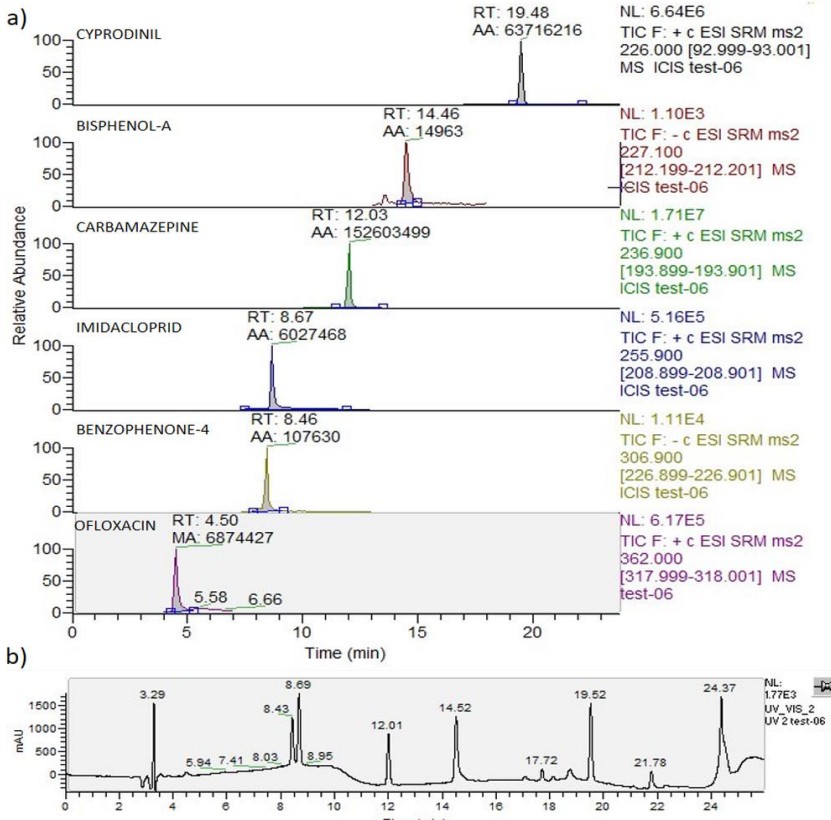

**Figure 3.** HPLC-UV-MS/MS analysis of the mixture at 0.1 mg/L. (**a**) individual ion current of each contaminant; (**b**) UV chromatogram at λ = 285 nm.

In Table 4, the average values of the pollutants' concentration in treated and untreated mixture are reported and compared with the respective LOD values.

**Table 4.** Concentration of the pollutants before (T = 0) the treatment and in the treated samples.

| Time (h) | Cyprodinil mg/L ± SD | Bisphenol-A mg/L ± SD | Carbamazepine mg/L ± SD | Imidacloprid mg/L ± SD | Benzophenone-4 mg/L ± SD | Ofloxacin mg/L ± SD |
|---|---|---|---|---|---|---|
| *LOD→* | (25 ng/L) | (0.1 mg/L) | (25 ng/L) | (25 ng/L) | (5 μg/L) | (25 ng/L) |
| 0 | 0.225 ± 0.003 | 0.535 ± 0.014 | 0.561 ± 0.015 | 0.500 ± 0.011 | 0.385 ± 0.002 | 0.640 ± 0.003 |
| 1 | <LOD | <LOD | <LOD | 0.433 ± 0.005 | <LOD | 0.012 ± 0.002 |
| 2 | – | – | – | 0.419 ± 0.002 | – | 0.009 ± 0.004 |
| 4 | – | – | – | 0.356 ± 0.007 | – | <LOD |
| 6 | – | – | – | 0.303 ± 0.005 | – | – |
| 8 | – | – | – | 0.284 ± 0.005 | – | – |

After 1 h of treatment, most of the parent molecules disappeared; the concentrations were under the limit of determination by the mass spectrometry analysis. This was an interesting result, since the concentration of each compound was about 0.5 mg/L, definitively higher than those present in wastewater or environmental samples. Conversely, 4 h of treatment were necessary to obtain an ofloxacin concentration lower than its LOD, confirming the high stability of the fluoroquinolone antibiotics and their long persistence once dispersed in the environment [43].

The most recalcitrant compound, however, was the insecticide, Imidacloprid. At the end of treatment, its concentration decreased only by 40%. This insecticide was recently banned in the European Union [44], but its resistance to degradation accounts for the concern about its accumulation in the environment, soil, and surface water.

### 3.2.2. The Organic Load

The concentration of the organic material in the effluent samples was determined as the COD and the TOC values. Both parameters showed little differences before and after the treatment. These data (Table 5) demonstrate that even the parent molecules were degraded, their oxidation products were still present, and negligible mineralization occurred.

**Table 5.** The COD and the TOC values in secondary effluent treated and untreated samples.

| Samples | COD (mg/mL) (±Uncertainty) | TOC (mg/L) |
|---|---|---|
| Untreated | 19.24 ± 5.12 | 21.6 ± 2.40 |
| Treated 6 h | | 14.6 ± 1.80 |
| Treated 8 h | 15.57 ± 1.52 | |
| | ΔCOD = 21.1 ± % | ΔTOC = 32.4 ± 2.2% |

### 3.2.3. The Biotoxicity Assays: The Bioluminescent Bacteria

The biotoxicity test based on the inhibition of the *Vibrio fisheri* light emission has been widely used to determine the presence of toxic compounds in liquid or solid samples by this rapid screening test. Bacterial bioluminescence is an energy-consuming phenomenon and light is emitted only when the organisms are in their best metabolic and physiologic conditions. Any component in the environment injuring the bacterial integrity or functions will produce a reduction or the disappearance of the light emission in a way directly proportional to the intensity of the suffered damage.

As expected, the mixture of the six pollutants was highly toxic to the luminescent bacteria reducing the light emission intensity to zero. The ozone treatment was effective in the cleavage of the parent molecules, leading to a partial restoration of luminescence emission, as shown in Figure 4. Prolonged-time treatment seems to increase and not reduce the toxicity of the mixture. Probably, the degradation process produces smaller

fragments able to interact more effectively with bacterial physiology. The increased toxicity of pollutants after degradation treatment is a well-known phenomenon. Figure 4 shown that the chronic toxicity assay can offer more precise information with respect to the acute one. After one hour, the untreated sample looked very similar to the control, but after 24 h the situation was completely different. The sensitivity of the assay is underlined by the fact that its results indicate that even the tap water, employed to prepare the solution, produced a 20% of light emission inhibition with respect to the control, probably due to the sanitizing treatment residues.

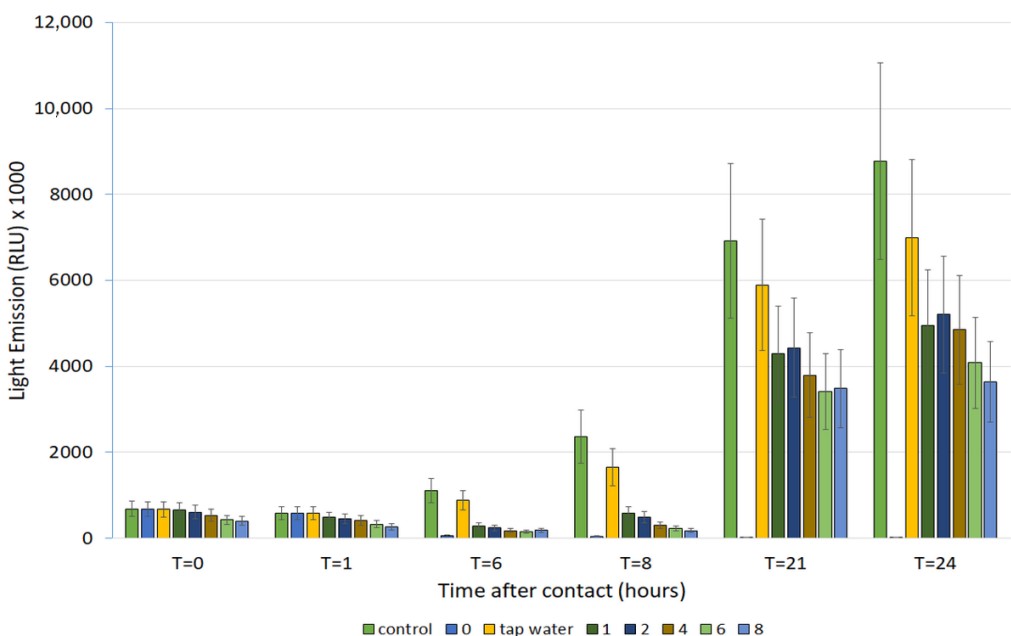

**Figure 4.** Light emission of luminescent bacteria in contact with the untreated (0) and the treated (1, 2, 4, 6, 8) mixture of pollutants. T = 0 was the light emission immediately after the microplate was prepared. T = 1 (6, 8, 21, 24) was the light emission after 1 (6, 8, 21, 24) h the microplate was prepared, i.e., the "Time after contact" (between bacteria and sample). T = 1 corresponds to the acute toxicity and T = 24 to the chronic toxicity.

By testing the effluent samples, no significant toxicity of both untreated and treated samples was revealed. On the other hand, in the chronic toxicity assays, treated samples frequently showed a light emission higher than that of the control. This was not surprising since WW was rich in organic compounds and the ozone degradation process produced smaller molecules acting as nutrients. The eutrophic effect produced by these nutrients must be carefully considered when planning the discharge of treated effluents in water bodies. Figure 5 reports the emissions of *Vibrio fisheri* in presence of the effluent samples.

On some occasions, the results were quite different: some samples were more toxic than expected or were greatly stimulated the light emission. Both effects were present in the group of samples whose data are shown in Figure 6. These samples were collected according to the new schedule, which included sampling after 15 and 30 min. The 30- and 120 min treated samples resulted heavily inhibited. The untreated and the treated samples collected at different times showed a stimulated light intensity. The assay on this group of samples was repeated various times to exclude an artefact. Actually, the chemical composition of the effluent can be different at each sampling and its effects on living organisms can be unpredictable, but the biotoxicological data will be, nevertheless, indispensable warning tools.

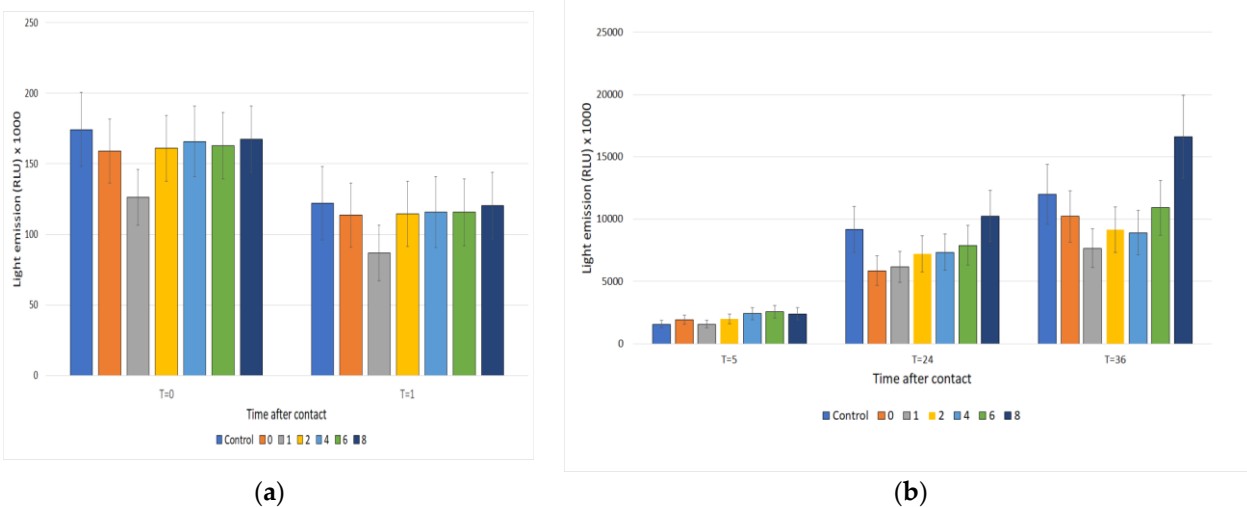

<div style="text-align: center;">(<b>a</b>)          (<b>b</b>)</div>

**Figure 5.** Biotoxicity assay of effluent samples treated for 0, 1, 2, 4, 6, and 8 h with ozone. (**a**) Acute toxicity, after 1 h of contact of the samples with the bioluminescent bacteria; (**b**) chronic toxicity, after longer contact time (24 or 36 h). The light emission values at T = 5 were included to emphasize the significant growth over time of bacteria in contact with the samples, which resulted in slight toxicity.

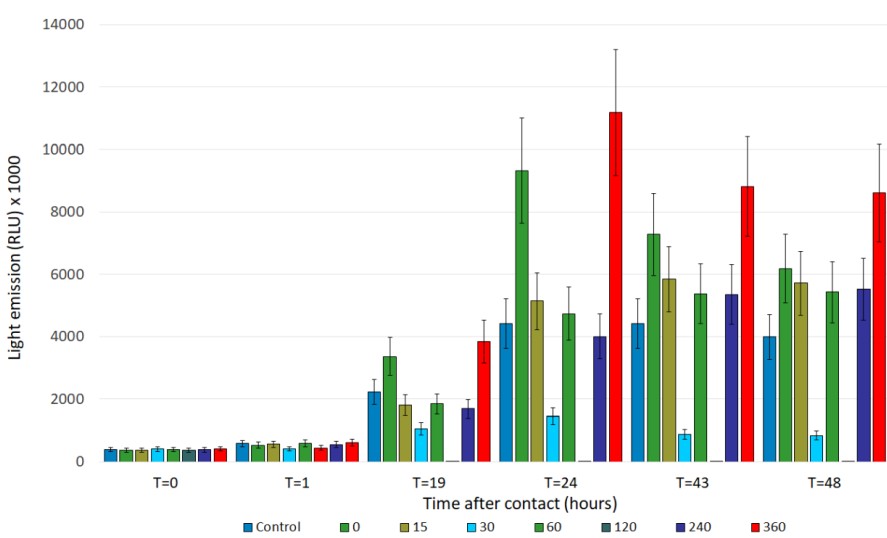

**Figure 6.** Light emission intensity of effluent samples collected according to the new schedule. The samples collected after 30 min and after 120 min of treatment were, surprisingly, highly toxic. Less surprising was the intensity of stimulation produced by the untreated sample.

### 3.2.4. The Biotoxicity Assays: The Green Algae

The food chain is founded on unicellular green algae; thus, knowing their response to the presence of pollutants in water bodies is of paramount importance. The sensitivity of the various strains to xenobiotics can differ greatly: the marine strains are usually more sensitive than the freshwater ones. The tap water produced a 10% growth inhibition of *D. tertiolecta* as previously observed in the assay with *V. fisheri*, another marine organism. The untreated sample produced a 97% reduction of the algal growth and all the treated samples no growth. The effects on the growth of *R. subcapitata* were just a little less important, as shown in Figure 7. These data indicated that the ozone treatment was not sufficient to remove all components responsible for negative effects on algae vitality, at least at the high concentrations present in the mixture.

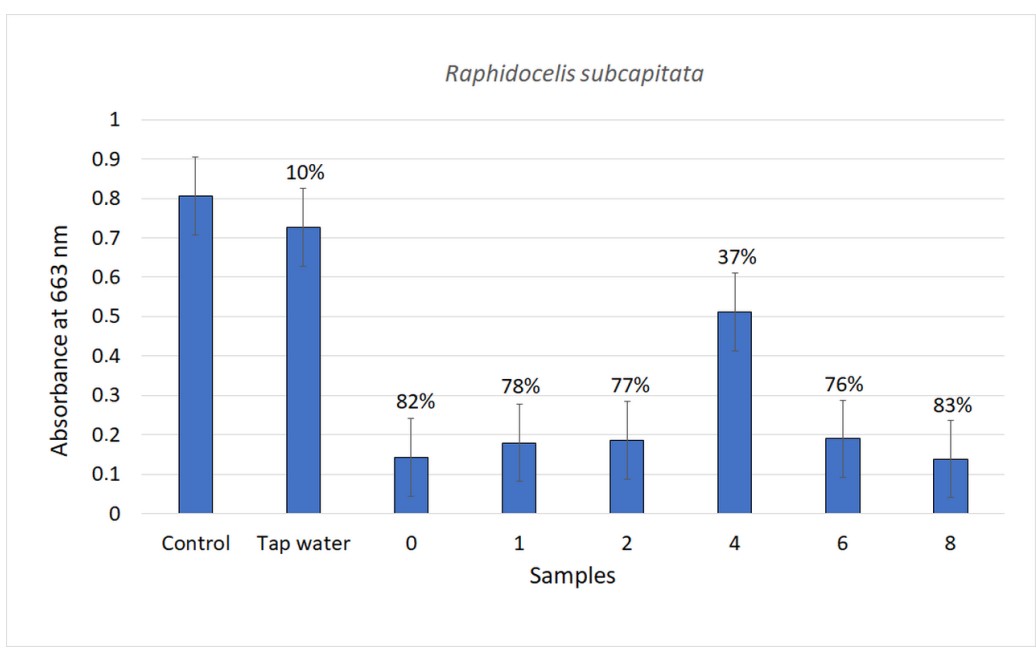

**Figure 7.** Chlorophyll absorbance quantified the growth rate of freshwater alga in contact with the samples of pollutant mixture treated by ozone. Above each sample is reported the % inhibition with respect to the Control. (0–8 are the hours of treatment).

The behavior of the two strains was different also while testing the effluent samples. The marine alga *D. tertiolecta* suffered a significant growth inhibition by the untreated and short-time treated samples. A slow recovery was observed in parallel with the treatment time increase. The growth of the freshwater alga *R.subcapitata* was equally inhibited by untreated sample, but the recovery was more rapid and less regular (Figure 8). The need for the larger possible set of biotoxicological tests to evaluate correctly the toxicity of complex samples is more than evident, since each kind of organism has a unique reaction to the presence of the same xenobiotic.

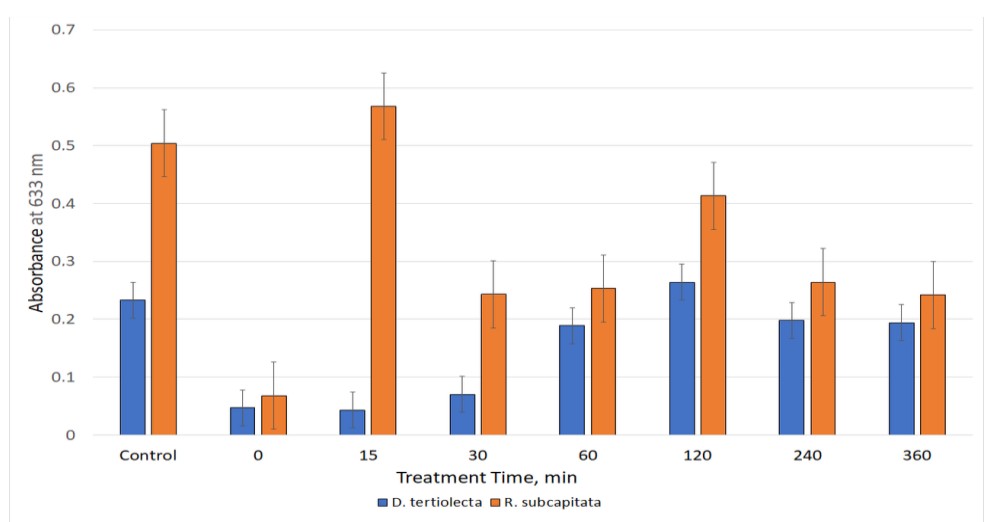

**Figure 8.** The histogram reported the absorbance values of chlorophyll in the cultures of *D. tertiolecta* and *R. subcapitata*. in contact with the effluent samples, treated or not (0), and compared with the respective Control, pure culture of algae.

*3.3. The Heterotrophic Plate Counts*

The main application of ozone is in the sanitization of rooms, surfaces, devices, foods, or solutions. To evaluate the effectiveness of the "Poolsan" device in sanitizing the sec-

ondary effluent samples we plated both the freshly treated samples and the same samples 7 days after. The heterotrophic plate count of thermophile bacteria was employed as the screening test for sanitization capacity. The effects of ozone on the number of CFUs in samples tested immediately after treatment and data about the possible regrowth of surviving bacteria were summarized in Table 6.

**Table 6.** The heterotrophic plate counts of freshly treated wastewater samples and of the same samples plated 7 days later. Regrowth phenomena did not occur, no colony was growing on plates of treated samples.

| Samples | CFU/100 mL $\pm$ Uncertainty | Regrowth CFU/100 mL |
|---|---|---|
| Untreated | 14,600 $\pm$ 4000 | — |
| $O_3$ treatment: 15 min | 1300 $\pm$ 600 | 0 |
| 30 min | 800 $\pm$ 300 | 0 |
| 60 min | 300 $\pm$ 100 | 0 |
| 120 min | 100 $\pm$ 40 | 0 |
| 240 min | 100 $\pm$ 50 | 0 |
| 360 min | 80 $\pm$ 30 | 0 |

## 4. Discussion

In this study, we investigated the performance of a self-assembled device to treat different kinds of wastewater by ozone injected into the water stream. $O_3$ dissolved in water has been used for sterilization and detoxification of fluids in all kinds of contexts [14,45–47], rarely alone but usually as a pre or post treatment, according to the chemical/biological content and to the final use of the reclaimed water.

Our aim was to test the possibility of employing our system both to treat small-moderate volumes of wastewater produced locally during various activities, such as agriculture, small industries, isolated communities, and to sanitize surface water or groundwater in areas where no potable water utilities are present, or they are temporarily not available because of some kind of emergency.

Concerning the removal of chemical pollutants from WW, the experiments on a mixed solution of six representatives of the so-called micropollutants or emerging pollutants at high concentration like in industrial WW gave positive results. The system was able to completely degrade most of the parent molecules after 1 h of treatment and this duration would likely be shorter after future optimization. Only one compound retained more than 50% of its initial concentration. The effectiveness of experimental system in the degradation of organic pollutants was further confirmed by the comparison with similar, already published data [2,31,35,48].

The COD and TOC values indicated that the overall organic content was not greatly changed after treatment, but the known role of ozone and its radicals is just to start the molecular degradation by oxidative attacks, enabling an easier biodegradation process, not to obtain extensive mineralization of the molecules [3,49,50].

To identify the degradation products was beyond our interest as several studies have been already published on the derivatives obtained by oxidation of various compounds included in our solution [2,3,34,51,52]. Moreover, we aimed to replicate real cases of WW treatment, in which a detailed identification on all the compounds and their derivatives is usually not feasible, and the treatment's effects are evaluated by simple tests, the biotoxicological or microbiological ones.

The evaluation of the treatment's effect on the micropollutants biotoxicity by the luminescent test was focused on the long-term results, since short-term assays can result in unrealistic data. As confirmed by data reported in Figure 4, the real toxicity of the untreated solution can be observed only in long-term measurements. The green algae response was also evaluated in long-term assays, to ensure the full expression of the pollutants' effects.

The results can be considered consistent with those obtained from bacteria, even though the sensitivity of these organisms was different. The marine algae, which was more sensitive, did not grow at all, while the freshwater algae was able to grow, but only about the 20% the control did, without significant differences among the samples. These differences among the various test organisms are the rule and each research work will define a different order of sensitivity for the same organisms tested on different analytes [53].

Beyond the metabolic differences among the organisms, the controversial effects of degradation procedures on organic pollutants' biotoxicity must be taken into consideration; often the degradation products are more toxic than the parental molecule [54].

In a few cases, the samples of municipal, post-precipitation plant effluent produced toxic effects; this frequently resulted in a growth stimulation both on bacteria and algae. The municipal WWTP effluents, containing not more than few µg/L of toxic micropollutants are rich in nutrients which can help the growth of living organisms [3]. This information is both significant and negative; an effluent containing a lot of nutrients will produce eutrophication phenomena in the receiving water bodies.

The second effect expected from ozone treatment, the sanitization of water, was evaluated by the non-selective heterotrophic plate count assay and although we could not identify the specific strains, these preliminary results were satisfying. In the freshly treated samples, the number of CFUs was reduced according to the treatment time, but the results from the same samples plated after 7 days were surprising: no one CFU was present on the plates. This feature, if it can be confirmed and accompanied by specific microbiological tests, would indicate that treated wastewater can stay sanitized over time. This finding is consistent with various previous works dealing with conditions for wastewater disinfection [55–57].

## 5. Conclusions

The present study was a preliminary work, aiming to ascertain the potentialities of the continuous-flow device in improving the various aspects of water quality. The next step of our research will be the investigation of the actual inactivation of typical bacterial strains such as fecal coliforms and other pathogenic and/or particularly resistant organisms. These data are necessary to confirm the efficacy of the treatment. When possible, the analytical evaluation of organic contaminants and degradation products in WW samples will be carried out.

Nevertheless, it is possible to confirm that the degradation rates of several very dangerous molecules, at unusually high concentrations after 1 h of treatment, were interesting, and the encouraging results suggest that the lower concentrations of the same or similar compounds present in real samples can be easily degraded in shorter time.

Following this first test on the system, we are planning to optimize its design, obtaining the best performance in the different situations in which it can be employed. In the continuous flow system, the blender tower size and the flow speed can be tailored to the water volume to be treated, on the final use of the treated water, and on the initial organic content. The amount of ozone and the treatment time can be recalculated according to the biological and chemical characteristics of the fluid under treatment, optimizing the costs and working time.

The development of such a simple, extremely flexible, and effective system can offer an novel way to recycle WW produced in small communities, during limited agricultural or industrial activities, or to improve drinking water quality in developing areas, small communities, and emergency situations.

On-site sanitization and remediation of small volumes of wastewater and not potable water sources using economic and easily conveyable systems is as important as remediating and reusing great volumes of wastewater produced by municipal and industrial treatment plants. The only alternative is to dissipate the wastewater, contaminating the environment at the same time, or risk human health through the consumption of unsafe drinking water

**Author Contributions:** Conceptualization, B.E., G.S., F.P. and E.N.F.; methodology, M.C.; validation, R.G. and L.B.; formal analysis, F.R.; investigation, B.E., M.L.N., L.B. and G.L.; data curation, M.L.N. and R.G.; resources, G.S. and F.P.; writing—original draft preparation, E.N.F.; writing—review and editing, E.N.F., B.E., M.C. and F.R. All authors have read and agreed to the published version of the manuscript.

**Funding:** This research received no external funding.

**Institutional Review Board Statement:** Not applicable.

**Informed Consent Statement:** Not applicable.

**Data Availability Statement:** The data presented in this study are available on request from the corresponding author with the permission of the company involved in the study, MET srl.

**Acknowledgments:** The author acknowledge the staff of MET srl for the kind supply of the Poolsan system and their technical support.

**Conflicts of Interest:** The authors declare no conflict of interest.

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
