# Peer review of "Safe Reuse of Wastewater: Organic Contaminants Degradation and Sanitization by Ozone in a Modulable Continuous-Flow System"

_applsci, doi:10.3390/app13127087_

Round 1

Reviewer 1 Report

This is an interesting manuscript on an important matter. Please see the following comments

1) The abstract contains some textbook information which should not be in an abstract. The abstract should contain information most pertinent to the research and the results found here

2) the quality of english language should be improved. for example

replace

Fresh water resources are globally under pressure by the increasing pop40 ulation, its increasing activities, and by the drought affecting day by day a 41 larger number of countries [1,2]. 

with

Fresh water resources are globally under pressure due to the increasing world pop40 ulation, the various anthropogenic activities, and the drought incidents that affect day by day a 41 larger number of countries [1,2]. 

you state

 To decrease the fresh water demand and the 42 wastewater production will be crucial both for the water availability, just now 43 and in the future, and for humans’ impact to the environment.  unclear meaning please rephrase

replace

The recycle and 44 reuse of any kind and volume of wastewater seems nowadays one of the im45 mediately feasible ways to cope with water scarcity, but in order to provide water which is safe for civil use disinfection and removal of pollutants are man47 datory procedures in order to avoid undesirable ecological and human health 48 effects, such as freshwater contamination, waterborne diseases, and ingestion 49 of toxic or carcinogenic compounds. Besides wastewaters of different quality 50 and origin, surface water and groundwater can be equally contaminated by mi51 croorganisms and chemical compounds, because of anthropologic activities or 52 natural conditions, then requiring the same adequate treatments before to be 53 considered safe for human consumption [3-5].

with

The recycle and 44 reuse of any kind and volume of wastewater seems nowadays one of the im45 mediately feasible ways to cope with water scarcity,. However  in order to provide water which is safe for use by humans disinfection and removal of pollutants are mandatory  in order to avoid undesirable ecological and human health 48 effects. Besides wastewaters of different quality 50 and origin, surface water and groundwater can be equally contaminated by mi51 croorganisms and chemical compounds, because of anthropologic activities or 52 natural conditions, thus requiring the same adequate treatments before to be 53 considered safe for human consumption [3-5].

there are many more mistakes which should be rectified, please consult with a fluent English speaker

3) there are also mistakes due to carelessness eg Fig 8 cannot be seen in the pdf version

4) the introduction is quite long and it contains some information on AOP that is better fit to the discussion, please shortent introduction to 2/3 of current length.

5) in the end of the introduction you should add a paragraph that explains the problem, shows what your research will be, shows what you are planning to achieve and why this is important for an international audience

6) in the materials and methods replace 2.1 The samples. with 2.1 Experimental set-up

7) in the materials and methods use passive voice instead of active voice throughout

8) in the materials and methods have a separate paragraph entitled Chemicals and regents where you write all the reagents you used together with manufacturer, city, country of origin

9) in the materials and methods for all the equipment used ultrasound, HPLC,  ozone generator, flasks, microscope, spectrophotometer etc give manufacturer, city, country of origin

10) for the WWTP wastewater there is very little information given. at least the population equivalents and the general outline of the WWTP should be given

11) all the data that were not produced by you but eg were provided by the WWTP company SHOULD NOT BE in the results but rather in the materials and methods section together with a short reference on the methods the company used to obtain these results. In any case, you need to link these results to yours in othe words how are these results helpful for your results? if they are not relevant to your results and discussion then they should be erased

12) the methods are not dscribed in detail. for all methods you applied you either have to give the protocol in detail or you have to refer to a PUBLISHED method. For 2.6 The heterotrophic plate count you do not decsribe the method (only the preparation) and you do not give a relevant reference and the same about 2.4 The samples organic load

13) I have great issues with the biotoxicity assays for example the algal inhibition assay. there are well known bioassays protocols by OECD, does your assay follow an OECD protocol? if so then you should mention the protocol and say only the modification you did, if not why you did not follow the OECD protocol since they are well established protocols?

14) the formula on page 6 should be rewritten using microsoft word formulas tool

15) I cannot really understand what is depicted at figs 4 to 6. what are the different groups? they are the pollutants mixture treated eg for 2 h after 2 h also the mixture treated for 2 h after 4 h time (4h from what time point?) this should be described in the legend of each figure. Since you have mean values and standard deviations you have to perform statistics (eg an one-way ANOVA between all the groups and the control with some post-hoc tests to see where the differences are and denote the differences with different letters eg a, b c, etc on each column)

16) the manuscript even though ot has so many data has a very short discussion and many of the matters found in the results are not discussed in length in the discussion. It is also very important to discuss more the ecotoxicity aspects on wastewater monitoring

please see and refer to the following

 Kungolos et al (2015) Assessment of vulnerable freshwater ecosystems and various aquatic effluents by means of ecotoxicological assays, Desalination and Water Treatment, 54:8, 2120-2129, DOI: 10.1080/19443994.2014.933617

Ortiz de García et al Ecotoxicity and environmental risk assessment of pharmaceuticals and personal care products in aquatic environments and wastewater treatment plants. Ecotoxicology.  doi: 10.1007/s10646-014-1293-8. 

Ma XY et al Ecotoxicity comparison of organic contaminants and heavy metals using Vibrio-qinghaiensis sp.-Q67. Water Sci Technol. 2013;67(10):2221-7. doi: 10.2166/wst.2013.113. PMID: 23676391.

please see comments above

Author Response

This is an interesting manuscript on an important matter. Please see the following comments

We are grateful to the Reviewer for his positive evaluation of our work and for his comments which were very helpful in revising and improving the text.

  • The abstract contains some textbook information which should not be in an abstract. The abstract should contain information most pertinent to the research and the results found here

We revised the Abstract deleting the unnecessary information

2) the quality of english language should be improved. for example

replace

Fresh water resources are globally under pressure by the increasing population, its increasing activities, and by the drought affecting day by day a 41 larger number of countries [1,2]. 

with

Fresh water resources are globally under pressure due to the increasing world population, the various anthropogenic activities, and the drought incidents that affect day by day a larger number of countries [1,2]. 

This sentence was deleted

 you state

 To decrease the fresh water demand and the wastewater production will be crucial both for the water availability, just now and in the future, and for humans’ impact to the environment.  unclear meaning please rephrase

replace

The recycle and  reuse of any kind and volume of wastewater seems nowadays one of the immediately feasible ways to cope with water scarcity, but in order to provide water which is safe for civil use disinfection and removal of pollutants are mandatory procedures in order to avoid undesirable ecological and human health effects, such as freshwater contamination, waterborne diseases, and ingestion of toxic or carcinogenic compounds. Besides wastewaters of different quality and origin, surface water and groundwater can be equally contaminated by microorganisms and chemical compounds, because of anthropologic activities or 52 natural conditions, then requiring the same adequate treatments before to be 53 considered safe for human consumption [3-5].

with

The recycle and 44 reuse of any kind and volume of wastewater seems nowadays one of the im45 mediately feasible ways to cope with water scarcity. However,  in order to provide water which is safe for use by humans disinfection and removal of pollutants are mandatory  in order to avoid undesirable ecological and human health 48 effects. Besides wastewaters of different quality 50 and origin, surface water and groundwater can be equally contaminated by mi51 croorganisms and chemical compounds, because of anthropologic activities or 52 natural conditions, thus requiring the same adequate treatments before to be 53 considered safe for human consumption [3-5].

            The first part of this sentence was deleted, the second one revised according to your suggestion.

there are many more mistakes which should be rectified, please consult with a fluent English speaker

            We agree with the reviewer, the manuscript underwent an extensive revision of the English language use.

  • there are also mistakes due to carelessness eg Fig 8 cannot be seen in the pdf version

We realized too late the problem created in the pdf version by Fig. 8, we will check more carefully the revised version.

  • the introduction is quite long and it contains some information on AOP that is better fit to the discussion, please shortent introduction to 2/3 of current length.

We definitely reduced the Introduction length.

  • in the end of the introduction you should add a paragraph that explains the problem, shows what your research will be, shows what you are planning to achieve and why this is important for an international audience

We add such a sentence at the end of the Introduction (lines 95-97)

  • in the materials and methods replace 2.1 The samples. with 2.1 Experimental set-up

This change was made.

  • in the materials and methods use passive voice instead of active voice throughout

We made this change throughout the M&M section.

  • in the materials and methods have a separate paragraph entitled Chemicals and regents where you write all the reagents you used together with manufacturer, city, country of origin

We create the new paragraph “Chemical and reagents” as suggested.

  • in the materials and methods for all the equipment used ultrasound, HPLC, ozone generator, flasks, microscope, spectrophotometer etc give manufacturer, city, country of origin

All the equipment were reported together with the information usually supplied in scientific papers.

  • for the WWTP wastewater there is very little information given. at least the population equivalents and the general outline of the WWTP should be given

This information was added at page 3, lines 113-125.

  • all the data that were not produced by you but were provided by the WWTP company SHOULD NOT BE in the results but rather in the materials and methods section together with a short reference on the methods the company used to obtain these results. In any case, you need to link these results to yours in othe words how are these results helpful for your results? if they are not relevant to your results and discussion then they should be erased

We decided to delete completely this set of data from the manuscript.

  • the methods are not dscribed in detail. for all methods you applied you either have to give the protocol in detail or you have to refer to a PUBLISHED method. For 2.6 The heterotrophic plate count you do not decsribe the method (only the preparation) and you do not give a relevant reference and the same about 2.4 The samples organic load

Concerning the biotoxicity assays we added the reference to the respective ISO procedure and we completed the description of the heterotrophic plate count assay. Concerning the organic load, the complete information about the two kits, employed according to the manufacturer’s instruction, is sufficient to the reader to understand which method was used.

  • I have great issues with the biotoxicity assays for example the algal inhibition assay. there are well known bioassays protocols by OECD, does your assay follow an Organisaton Economical Cooperation Development protocol? if so then you should mention the protocol and say only the modification you did, if not why you did not follow the OECD protocol since they are well established protocols?

The laboratory performing the biotoxicity assay is an Analytical Chemistry Lab and they followed since long time the protocols reported in various ISO protocols, especially those concerning the Water Quality, employed at international level and currently valid.

14) the formula on page 6 should be rewritten using microsoft word formulas tool

            The formula was completely rewritten

15) I cannot really understand what is depicted at figs 4 to 6. what are the different groups? they are the pollutants mixture treated eg for 2 h after 2 h also the mixture treated for 2 h after 4 h time (4h from what time point?) this should be described in the legend of each figure. Since you have mean values and standard deviations you have to perform statistics (eg an one-way ANOVA between all the groups and the control with some post-hoc tests to see where the differences are and denote the differences with different letters eg a, b c, etc on each column)

We agree completely with this comment, the legends and the text were not detailed enough to result clear for who was not involved in the work. Now we improved the legends and the text, explaining exactly the meaning of the different groups. We know about the possibility to perform statistics when means and SD are available, but this kind of treatment will not add any information to our biotoxicological data. They were necessary to ascertain the positive effect, and/or to exclude a negative one, of ozone treatment on the toxicity of pollutants and we obtained this information.  

16) the manuscript even though ot has so many data has a very short discussion and many of the matters found in the results are not discussed in length in the discussion. It is also very important to discuss more the ecotoxicity aspects on wastewater monitoring

The Discussion was revised and a Conclusion paragraph was added. We added some more comments on data also along the text. Nevertheless, in our opinion this work was not focused on wastewater monitoring or exactly on ecotoxicological aspects, rather on the effectiveness of a new system for water treatment. Probably, in the next future, we will investigate deeper its effects from the ecotoxicological point of view but this will require a larger set of biological data.

please see and refer to the following

 Kungolos et al (2015) Assessment of vulnerable freshwater ecosystems and various aquatic effluents by means of ecotoxicological assays, Desalination and Water Treatment, 54:8, 2120-2129, DOI: 10.1080/19443994.2014.933617

Ortiz de García et al Ecotoxicity and environmental risk assessment of pharmaceuticals and personal care products in aquatic environments and wastewater treatment plants. Ecotoxicology.  doi: 10.1007/s10646-014-1293-8. 

Ma XY et al Ecotoxicity comparison of organic contaminants and heavy metals using Vibrio-qinghaiensis sp.-Q67. Water Sci Technol. 2013;67(10):2221-7. doi: 10.2166/wst.2013.113. PMID: 23676391.

Reviewer 2 Report

Title: Safe Reuse of Wastewaters: Organic Contaminants Degradation and Sanitization by Ozone in a Modulable Continuous-Flow System by Esposito et al.

The authors investigated the efficacy of a mobile ozone module for wastewater treatment. After determining that ozone effectively degrades pharmaceuticals present in the wastewater, they focused on studying how the degradation products interact with the microflora, specifically whether they are toxic to living organisms. The results may indicate whether ozonated wastewater can be further treated biologically.

The research was well-planned and executed, but the structure of the paper deviates from common conventions. The reviewer acknowledges that their own opinions may influence this assessment.

In the ‘Introduction’, the authors reiterate widely known views that do not contribute anything new to the knowledge and topic of the reviewed work. At least the first two sentences in the ‘Introduction’ could be removed.

Lines 108-116 describe the aim of the study and should be placed at the end of the ‘Introduction’.

Lines 117-128 should be moved, after some revisions, to the ‘Conclusions’ section, which is currently missing.

Lines 163-165 describe potential future expansions of the system, rather than the current system, and should be removed from this section.

Line 213 - mg/l should be changed to mg/L.

Line 235 -   +/-SD introduces ambiguity in meaning. It is also unclear whether in the Table 3 and other (7.87±0.11, 7.43±0.10), ±0.11 represents SD (standard deviation) or ‘uncertainty’ which is typically written after the ± sign. If the authors use 7.87±SD, 7.43±SD in the Tables, they should indicate this in the Table headers.

Table 4 - The authors should provide the meaning of the abbreviation CFU earlier, as most readers may not be familiar with bacterial culture terminologies. Additionally, mg/L is not the same as (CFU/100mL), so it is necessary to specify which unit applies to which results in that table.

Figure 2. Structural formulas of chemical compounds should be redrawn as on my printout there are gray stripes left after editing the grayscale images. The same applies to Figure 3.

Line 385 - The words ‘It is’ left alone after some mistake at editing.

Line 391 - for consistency, write 0.5 mg L-1 as 0.5 mg/L

Lines 401-406 - write this in the ‘Conclusions’ section.

Table 6 - Explain why the measured values of COD change (increase and decrease alternately) over time. Could this be an experimental error? How many replicates were performed? Shouldn't the most deviating results be rejected?

Figure 5 and Figure 6 - I suggest using MRLU or kRLU on the ‘Light emission’ axis.

Figure 8 - is only partially visible.

Table 7 - Explain the meaning of ‘ASPETTARE’ or translate it into English.

Line 559 - Change ‘device’ to ‘system’.

Lines 568-574 - should be in the ‘Materials and methods’ section.

The ‘Discussion’ section contains content that should be in the ‘Conclusions’ section.

Line 603 – ‘but different analyte [63]’, however, in the ‘References’ section, the last citation is numbered 62, which is not cited in the text.

Lines 618 and onwards should already be in the ‘Conclusions’ section.

The number of cited articles is too high. References 30 and 58 may not be related to the presented research.

 The paper requires significant changes in its structure, and in its current version, it is not suitable for publication.

Author Response

Title: Safe Reuse of Wastewaters: Organic Contaminants Degradation and Sanitization by Ozone in a Modulable Continuous-Flow System by Esposito et al.

The authors investigated the efficacy of a mobile ozone module for wastewater treatment. After determining that ozone effectively degrades pharmaceuticals present in the wastewater, they focused on studying how the degradation products interact with the microflora, specifically whether they are toxic to living organisms. The results may indicate whether ozonated wastewater can be further treated biologically.

The research was well-planned and executed, but the structure of the paper deviates from common conventions. The reviewer acknowledges that their own opinions may influence this assessment.

In the ‘Introduction’, the authors reiterate widely known views that do not contribute anything new to the knowledge and topic of the reviewed work. At least the first two sentences in the ‘Introduction’ could be removed.

Agreeing with this comment, we revised the Introduction and delete the first two sentence

Lines 108-116 describe the aim of the study and should be placed at the end of the ‘Introduction’.

The aim of the study was placed quite at the end of the paragraph, since we had to introduce a short sentence after it, required by another Reviewer.

Lines 117-128 should be moved, after some revisions, to the ‘Conclusions’ section, which is currently missing.

            We added now the Conclusion section and this fragment was moved to it

Lines 163-165 describe potential future expansions of the system, rather than the current system, and should be removed from this section.

Line 213 - mg/l should be changed to mg/L.

            This typing mistake was corrected.

Line 235 -   +/-SD introduces ambiguity in meaning. It is also unclear whether in the Table 3 and other (7.87±0.11, 7.43±0.10), ±0.11 represents SD (standard deviation) or ‘uncertainty’ which is typically written after the ± sign. If the authors use 7.87±SD, 7.43±SD in the Tables, they should indicate this in the Table headers.

We apologize for this confusion in showing the data. In the revised manuscript the meaning of the values added after the ± sign.

Table 4 - The authors should provide the meaning of the abbreviation CFU earlier, as most readers may not be familiar with bacterial culture terminologies. Additionally, mg/L is not the same as (CFU/100mL), so it is necessary to specify which unit applies to which results in that table.

We decided to delete this Table.

Figure 2. Structural formulas of chemical compounds should be redrawn as on my printout there are gray stripes left after editing the grayscale images. The same applies to Figure 3.

            We are sorry, but we did not find any defects or stripes on the two Figures. Figure 3 was modified anyway.

Line 385 - The words ‘It is’ left alone after some mistake at editing.

This was a classical typing mistake slipped from last checking

Line 391 - for consistency, write 0.5 mg L-1 as 0.5 mg/L

            We change throughout the text mg L-1 to mg/L

Lines 401-406 - write this in the ‘Conclusions’ section.

            We moved this sentence to the Conclusion.

Table 6 - Explain why the measured values of COD change (increase and decrease alternately) over time. Could this be an experimental error? How many replicates were performed? Shouldn't the most deviating results be rejected?

Actually, these values do not look well, but the samples were done in triplicate. We hypothesize that the whole sample was not completely homogeneous and the various specimens could be different. Nevertheless, the most important information is the little reduction of COD and TOC values at the end of treatment. We decided to focus on this point, deleting the intermediate values.

Figure 5 and Figure 6 - I suggest using MRLU or kRLU on the ‘Light emission’ axis.

We modified the values on the y axis, indicated as (x1000).

Figure 8 - is only partially visible.

The Figure 8 disappeared in the pdf version; we will check more carefully this revised version.

Table 7 - Explain the meaning of ‘ASPETTARE’ or translate it into English.

            Now deleted, it was a note inserted during the manuscript preparation, probably hidden by a Figure and “moved” by the change to pdf version.

Line 559 - Change ‘device’ to ‘system’.

            We made this change.

Lines 568-574 - should be in the ‘Materials and methods’ section.

            We moved this part to the Materials and methods’ section

The ‘Discussion’ section contains content that should be in the ‘Conclusions’ section.

            The Discussion section was revised and a Conclusion section was added

Line 603 – ‘but different analyte [63]’, however, in the ‘References’ section, the last citation is numbered 62, which is not cited in the text.

            Again, this was a typing mistake; according to your suggestion, in the revised text the number of the references was reduced and now carefully checked

Lines 618 and onwards should already be in the ‘Conclusions’ section.

            We moved this part to the new Conclusion section.

The number of cited articles is too high. References 30 and 58 may not be related to the presented research.

            As mentioned before, we reduced the number of references, deleting not only the n.30 and 58.

 The paper requires significant changes in its structure, and in its current version, it is not suitable for publication.

            We are grateful to the Reviewer for his comments, very helpful in improving our manuscript and we hope we complied with his suggestions.

Reviewer 3 Report

Article entitled Safe reuse of wastewaters: Organic contaminants degradation and sanitization by ozone in a modulable continuous-flow system written by Biagio Esposito, Massimo Capobianco, Maria Luisa Navacchia, Giampaolo Sabia, Roberta Guzzinati, Francesco Riminucci, Luca Bolelli, Giorgio Longino, Elida N. Ferri and submitted to Applied Sciences journal as a draft no applsci-2360148 deals with an important issue of development of methods that allow for efficient micropollutants removal from wastewater and decrease toxicity.

The article is in journal’s scope. Therefore, it could be considered for publication in Applied Sciences. As English is not my native language, I am not able to assess language correctness. However, while reading, I found some statements missing, confusing or unclear. Below I enclose the list of my comments.

As ozone solubility is small and slow, depends on many factors, including thermodynamic ones eg.  pressure and temperature. I think that if possible, it is worth supplementing this data in the description of the experiment.

In the case of ozonation, the oxidation process takes place mainly on the surface of the bubbles migrating through the bed. Therefore, the absolute dose of ozone, although important, is not crucial. Bubble size and contact time are equally important. Did the Authors control these parameters? If so, it is worth supplementing this information in the article.

It is worth specifying the height of the reactor - it affects the time of contact with gaseous ozone and thus the solubility.

Was an ozone destructor used? Was the amount of ozone leaving the reaction chamber controlled?

I propose to subject figure 3 to graphic processing to give it a more "scientific" look. Similarly with other figures.

Has an attempt been made to identify the compounds formed during ozonation?

There are a few minor typos in the text.

The Authors write that they determined TOC. The range given by the Authors is 15-150mg/L. Meanwhile, the initial COD is around 18mg/L – table 4. COD can be expected to fall below the limit of quantification after a short period of time. Why wasn't a more sensitive method used?

Table 6. There seems to be a problem with the reliable determination of TOC in wastewater - Table 6. It is worth explaining how it is possible that TOC increases during the process. Do the Authors introduce any compounds during the process that may interfere with the TOC determination?

In table 6 there is no information about which the Authors write in line 416.

Figure 8 is completely illegible.

I have not found anywhere the results of the TOC determination, which the authors write about in the line 582.

Conclusion is missing.

Based on my comments and general impression I suggest major revision.

Author Response

Article entitled Safe reuse of wastewaters: Organic contaminants degradation and sanitization by ozone in a modulable continuous-flow system written by Biagio Esposito, Massimo Capobianco, Maria Luisa Navacchia, Giampaolo Sabia, Roberta Guzzinati, Francesco Riminucci, Luca Bolelli, Giorgio Longino, Elida N. Ferri and submitted to Applied Sciences journal as a draft no applsci-2360148 deals with an important issue of development of methods that allow for efficient micropollutants removal from wastewater and decrease toxicity.

The article is in journal’s scope. Therefore, it could be considered for publication in Applied Sciences. As English is not my native language, I am not able to assess language correctness. However, while reading, I found some statements missing, confusing or unclear. Below I enclose the list of my comments.

            We completely agree with this comment and we made a careful revision both of the text and of the English language use.

As ozone solubility is small and slow, depends on many factors, including thermodynamic ones eg.  pressure and temperature. I think that if possible, it is worth supplementing this data in the description of the experiment.

            The manufacturer of the system known very well the difficulties to have ozone in solution at useful concentrations; the system has a temperature control system which stop the generator when the temperature reaches 30°C. He did not control the pressure of the ozone flow but directly the amount of ozone actually in solution. This information is already present in the text.

In the case of ozonation, the oxidation process takes place mainly on the surface of the bubbles migrating through the bed. Therefore, the absolute dose of ozone, although important, is not crucial. Bubble size and contact time are equally important. Did the Authors control these parameters? If so, it is worth supplementing this information in the article.

We agree completely with the Reviewer on the importance of the bubbles size and of the contact time, well known fundamental parameters, but that described in our manuscript was a preliminary work to test the performance of the system in this new, specific application. The ozone flow created a microbubbles stream and the contact time was maintained constant by changing the blender volume according to the treated water volume. Further details were protected by the manufacturer. We plan to study in details these and other parameters during the future optimization of the system. 

It is worth specifying the height of the reactor - it affects the time of contact with gaseous ozone and thus the solubility.

In the case of this study, the reactor, or blender, was a container of 1 liter volume, useful to treat 50 L of water.

Was an ozone destructor used? Was the amount of ozone leaving the reaction chamber controlled?

            The generator has an ozone destructor, and the amount of leaving ozone was not determined. The technicians operating the system worked according to the regulations in force.

I propose to subject figure 3 to graphic processing to give it a more "scientific" look. Similarly with other figures.

All Figures in the manuscript were improved in quality and style.

Has an attempt been made to identify the compounds formed during ozonation?

The identification of the oxidation products was any case out of the scope of this work. On the other hand, the degradation of the pollutants in the mixture has been extensively studied, including the identification of their derivatives; the transformations occurring in a complex mixture such as WWTPS effluent could be investigated, but the unpredictable variations of its composition make useless such a information.  

There are a few minor typos in the text.

            The manuscript has been carefully revised and various typos corrected.

The Authors write that they determined TOC. The range given by the Authors is 15-150mg/L. Meanwhile, the initial COD is around 18mg/L – table 4. COD can be expected to fall below the limit of quantification after a short period of time. Why wasn't a more sensitive method used?

            The 15-150mg/L range is that of the employed Kit and we did not change it to another one in a lower range because we were able to obtain good measurement by using it. We already known that the limit of quantification would not change a lot after the ozone treatment.

Table 6. There seems to be a problem with the reliable determination of TOC in wastewater - Table 6. It is worth explaining how it is possible that TOC increases during the process. Do the Authors introduce any compounds during the process that may interfere with the TOC determination?

            In Table 6 we reported only the COD values. Actually, these values do not look well, but the samples were done in triplicate. We hypothesize that the whole sample was not completely homogeneous and the various specimens resulted a little different. No one compound was added during the process. In the revised version of the manuscript, we decided to focus on the small difference between the treated and untreated values, deleting the intermediate ones.

In table 6 there is no information about which the Authors write in line 416.

            The note “data not showed” was missed and, actually, no one information was in Table 6. Now we modified the Table adding the TOC values of the untreated and the 6 h treated samples.      

Figure 8 is completely illegible.

The Figure 8 “disappeared” in the pdf version; we will check more carefully in this revised version the position of all figures.

I have not found anywhere the results of the TOC determination, which the authors write about in the line 582.

In the line 582 we did not report data about the TOC determination, but data concerning the HPLC-MS measurement of parent molecules’ concentration in pollutants mixture after the ozone treatment.

Conclusion is missing.

In the revised version the Conclusion section has been added.

Reviewer 4 Report

The Introduction is too long for a well known technology as ozonation. Please shorten.

Line 82. Ozonation alone is not considered an AOP. Maybe it combination with other methods or oxidizers such as H2O2 or UV light, TiO2...

Line 117- . Please do not provide results here.

Table 4. I guess that concentrations are provided in mg/L, not mg/mL.

Fig. 8 is not visible.

Line 290. Please explain pH increment.

Line 311: "...and were released with the samples", what was released with the samples?

Results regardinn increased toxicity are typical: degradation intermediates can be more toxic than the parental ones. The authors do not discuss this point.

In general the English is good but there some errors and typos that should be corrected.

Author Response

The Introduction is too long for a well known technology as ozonation. Please shorten.

            According to this very useful suggestion we revised the Introduction and the too obvious and known information about ozonation were deleted.

Line 82. Ozonation alone is not considered an AOP. Maybe it combination with other methods or oxidizers such as H2O2 or UV light, TiO2...

            We agree completely with this comment. The effects of ozone alone cannot be considered as an AOP. Nevertheless, several Authors include, incorrectly, ozone in the list of AOP. In the present work we tested the effects of ozone alone, but it is possible that the future development of this research will involve “true” AOPs, the usual topic of our works.

Line 117- . Please do not provide results here.

            We deleted this part during the Introduction revision.

Table 4. I guess that concentrations are provided in mg/L, not mg/mL.

            We decided to delete Table 4.

Fig. 8 is not visible.

The Figure 8 “disappeared” in the pdf version; we will check more carefully in this revised version the position of all figures

.

Line 290. Please explain pH increment.

Actually, the increment was not so significant and it would be extremely difficult to propose a “chemical responsible” of pH increase in such a complex mixture of compounds like the WW is.

Line 311: "...and were released with the samples", what was released with the samples?

This part of the text was deleted during revision process.

Results regarding increased toxicity are typical: degradation intermediates can be more toxic than the parental ones. The authors do not discuss this point.

            The increase of toxicity after degradation treatments is a well-known, adverse effect. Moreover, this effect can appear with more or less intensity in the different test organisms, but to discuss this aspect was out of the main aim of this work. We just mentioned it at the lines 419-20.

In general the English is good but there some errors and typos that should be corrected.

            The manuscript underwent an extensive revision of the English language use and of typos, hopefully complete.

Round 2

Reviewer 1 Report

acceptable

please see comments

Author Response

Comments and Suggestions for Authors: acceptable

Comments on the Quality of English Language please see comments (moderate)

We thank the Reviewer for his positive evaluation. The English Language was revised again.

Reviewer 2 Report

Title: Safe Reuse of Wastewaters: Organic Contaminants Degradation and Sanitization by Ozone in a Modulable Continuous-Flow System by Esposito et al.

The research was well-planned and executed, but the format of the paper deviates from common conventions. Although the authors were repeatedly reminded to use passive voice, they only did so in some instances.

The "Introduction" section has been shortened, and the discussion of results has been removed from this part. However, the goal of the study is still not written in accordance with the conventions commonly accepted in such works.

Detailed comments:

Line 70 - "improves the color" can be ambiguous - it could mean either enhancing the color or, as the authors intended, removing the discoloration of the wastewater.

Lines 84-93 - rework this part. This section should describe what will be done in the study, not provide a description of what has already been done. Avoid using phrases like "we tested".

Line 101 - change "250 ml" to "250 mL".

Figure 2 - The structural formulas of chemical compounds should be redrawn as there are gray stripes left on my printout after editing the grayscale images. The same applies to Figure 3. The authors have stated in their response that there are no gray stripes, but if Figures 2 and 3 are printed on a laser printer, these gray artifacts are visible. On a computer screen in a pdf file, they are not visible. In Figure 2, this applies to three structural formulas, while in Figure 3, it applies to the scales, peaks, and all the side descriptions.

Line 365 - change "we report" to passive voice.

Line 376 - "confirming the high stability" - if the word "confirming" is kept, then a source should be added to inform about the "high stability". If the word "indicating" were used, then the source would not be necessary.

Figure 5 - align the bases of both figures and place the captions a) and b) on the respective figures - not below figures

Line 447 - change the sentence structure to "It must be mentioned that on some occasions, the results were quite different:" or remove "It must be mentioned that".

Line 467 - remove the double comma.

Line 486 - "maximum growth was produced by the 4-hour treated sample (Figure 8a)" - however, according to Figure 8, the maximum is reached after 120 minutes, i.e., after 2 hours. Additionally, there is no Figure 8a  8b. There is some confusion in this part that needs to be carefully checked.

Lines 518-527 - It is not clear whether these comments refer to this study or are a general consideration that should be in the Introduction.

Line 533 - instead of "our system," use "experimental system" or "built system".

Lines 542-547 - change active voice to passive voice - remove "our" - change to "built".

Line 550 - change "realistic information" to "realistic data".

Line 562 - add a source for the information that "often the degradation products are more toxic than the parent compounds".

Line 584 - rephrase the sentence.

Line 608 - maybe change "or not potable water" to "and not potable water".

In some places the language shows personal feelings to the system they built.

Author Response

Title: Safe Reuse of Wastewaters: Organic Contaminants Degradation and Sanitization by Ozone in a Modulable Continuous-Flow System by Esposito et al.

The research was well-planned and executed, but the format of the paper deviates from common conventions. Although the authors were repeatedly reminded to use passive voice, they only did so in some instances.

In this revised version we introduced the use of passive voice throughout the text.

The "Introduction" section has been shortened, and the discussion of results has been removed from this part. However, the goal of the study is still not written in accordance with the conventions commonly accepted in such works.

Detailed comments:

Line 70 - "improves the color" can be ambiguous - it could mean either enhancing the color or, as the authors intended, removing the discoloration of the wastewater.

The sentence was modified, removing the uncorrected term “improves”

Lines 84-93 - rework this part. This section should describe what will be done in the study, not provide a description of what has already been done. Avoid using phrases like "we tested".

The text was deeply revised, and phrases like “we tested” were deleted throughout the manuscript.

Line 101 - change "250 ml" to "250 mL".

This change was done

Figure 2 - The structural formulas of chemical compounds should be redrawn as there are gray stripes left on my printout after editing the grayscale images. The same applies to Figure 3. The authors have stated in their response that there are no gray stripes, but if Figures 2 and 3 are printed on a laser printer, these gray artifacts are visible. On a computer screen in a pdf file, they are not visible. In Figure 2, this applies to three structural formulas, while in Figure 3, it applies to the scales, peaks, and all the side descriptions.

We are sorry for the problem you encountered, but we made various way to visualize the Figures and they resulted always OK.

Line 365 - change "we report" to passive voice.

This change was done

Line 376 - "confirming the high stability" - if the word "confirming" is kept, then a source should be added to inform about the "high stability". If the word "indicating" were used, then the source would not be necessary.

We maintained the word “confirming” and we added the required source.

Figure 5 - align the bases of both figures and place the captions a) and b) on the respective figures - not below figures
The captions a) and b) were placed on the respective figures, now aligned.

Line 447 - change the sentence structure to "It must be mentioned that on some occasions, the results were quite different:" or remove "It must be mentioned that".

We removed "It must be mentioned that" from that sentence.

Line 467 - remove the double comma.

This change was done.

Line 486 - "maximum growth was produced by the 4-hour treated sample (Figure 8a)" - however, according to Figure 8, the maximum is reached after 120 minutes, i.e., after 2 hours. Additionally, there is no Figure 8a  8b. There is some confusion in this part that needs to be carefully checked.

We are grateful to the Reviewer for his comment. Actually, this sentence was not modified after we changed the type of Figure. Now this part is rewritten according to the new Figure 8.

Lines 518-527 - It is not clear whether these comments refer to this study or are a general consideration that should be in the Introduction.

These comments are useful to clarify the particular design of our experiments.

Line 533 - instead of "our system," use "experimental system" or "built system".

We changed the sentence employing the term “experimental”

Lines 542-547 - change active voice to passive voice - remove "our" - change to "built".

We did all the suggested changes.

Line 550 - change "realistic information" to "realistic data".

This change was done.

Line 562 - add a source for the information that "often the degradation products are more toxic than the parent compounds".

The required source was added.

Line 584 - rephrase the sentence.

The text was modified.

Line 608 - maybe change "or not potable water" to "and not potable water".

We made this change

Reviewer 3 Report

This is my second review of this article. The Authors answered all f my comments. Suggested correctiosn have been applied.  Second version is better than the first one. However, there are some typos in the article that change the substantive meaning. The entire article should be thoroughly checked. Examples of errors are e.g. line 208 (and others) Hack - should be Hach. Table 5 - COD should be in mg/L, not in mg/mL.

Author Response

This is my second review of this article. The Authors answered all f my comments. Suggested correctiosn have been applied.  Second version is better than the first one. However, there are some typos in the article that change the substantive meaning. The entire article should be thoroughly checked. Examples of errors are e.g. line 208 (and others) Hack - should be Hach. Table 5 - COD should be in mg/L, not in mg/mL.

We thank the Reviewer for his useful suggestions and comment. We corrected the mentioned errors and we revised carefully the text

Reviewer 4 Report

The manuscript can be accepted

Good in general but some minor errors are still present

Author Response

Comments and Suggestions for Authors    The manuscript can be accepted

Comments on the Quality of English Language:  Good in general but some minor errors are still present

We thank the Reviewer for his positive evaluation. The English Language was revised again